# The Role of Coping Styles in Mediating the Dark Triad and Bullying: An Analysis of Gender Difference

**DOI:** 10.3390/bs13070532

**Published:** 2023-06-26

**Authors:** Fangjing Xia, Mengjun Liu, Tour Liu

**Affiliations:** 1School of Psychology, Central China Normal University, Wuhan 430079, China; 2Key Laboratory of Adolescent Cyberpsychology and Behavior (CCNU), Ministry of Education, Wuhan 430079, China; 3Faculty of Psychology, Tianjin Normal University, Tianjin 300387, China; 4Key Research Base of Humanities and Social Sciences of the Ministry of Education, Academy of Psychology and Behavior, Tianjin Normal University, Tianjin 300387, China; 5Tianjin Social Science Laboratory of Students’ Mental Development and Learning, Tianjin 300387, China

**Keywords:** Dark Triad, bullying, coping styles, aggression, gender differences

## Abstract

Recently, the phenomenon of school bullying has gradually become a primary focus of social attention. To reduce the occurrence of bullying, it is important that we explore the psychological mechanisms of students with bullying tendencies. We conducted mediation models through a multi-group analysis to verify the mediating effect of coping styles on the relationship between dark personality traits and bullying, and further explored the differences in this mechanism between male and female groups. The participants were 772 high school students recruited from a middle school in Tianjin, China. They completed a set of self-reported measurements including the Dirty Dozen (DD), Simplified Coping Style Questionnaire (SCSQ), Buss–Perry Aggression Questionnaire (BPAQ), and Reactive–Proactive Aggression Questionnaire (RPQ). All the measurement instruments have acceptable reliability and validity. The results of the multi-group multiple models indicated that (1) there are significant gender differences in bullying behavior, with males scoring significantly higher than females. Moreover, the gender difference was only reflected in proactive aggression, in which males had higher scores than females; there was no significant gender difference in reactive aggression. (2) In the group of females, both negative and positive coping styles partially moderated the relationship between the Dark Triad and bullying. However, in the group of males, only negative coping styles partially mediated the relationship between the Dark Triad and bullying. (3) The above results also held for proactive aggression. In conclusion, our study highlights the gender differences in the mediating effect of coping styles on the relationship between the Dark Triad and bullying and proactive aggression. These findings contribute to better shared understanding of gender-related aspects in school bullying.

## 1. Introduction

Bullying is a common form of aggressive behavior that frequently occurs on campuses; it is perpetrated by individuals or groups with more power or strength than the weaker party [1]. According to data published by UNESCO in 2018 [2], 32% of students worldwide were bullied at least once within nearly a month of the investigation. Most previous research has focused on victims [3,4], but we know less about bullies; knowledge on bullies is essential for researching the psychological mechanisms of bullying. According to previous studies, teenagers with bullying behavior usually suffer from mental health problems [5], such as a higher incidence of mental illness, experiences of non-suicidal self-injury, a higher tendency to experience depression and anxiety, and the risk of antisocial personality disorder [6]. Therefore, more research needs to be carried out on the psychological mechanisms of bullying, as this is critical for intervention and prevention. 

The occurrence of bullying is influenced by multiple factors, such as empathy, self-esteem, and the quality of parental management [7,8,9]. Personality is recognized as one significant psychological factor among them. Many studies have explored the relationship between personality traits and bullying using the Big Five Personality Inventory. Bullying and cyberbullying are negatively correlated with agreeableness, positively associated with neuroticism, and somewhat negatively correlated with conscientiousness [10]. The meta-analysis of Mitsopoulou and Giovazolias [11] found that specific personality dimensions and variables could predict participation in bullying with convergence. In recent years, the “dark side” of the human personality has gradually received public attention [12], which makes up for the limitations of knowledge on only the traditional “bright side” of the human personality, which is made up of the Big Five personality traits. The Dark Triad is an antisocial personality trait group that has been paid attention by researchers over the years [13]. It is composed of three traits: Machiavellianism, psychopathy, and narcissism. Jain et al. [14] indicated that people with dark personality traits are more aggressive, which further reinforces the belief that bullying is a behavior used to assume a position of leadership among people [15]. Goodboy and Martin [16] found that the Dark Triad could positively predict cyberbullying, and people with such personality traits are more likely to attack others without provocation. It can be seen from these studies that Dark Triad may be an important predictor of bullying.

People with different personalities usually adopt different styles of coping with the external environment. Coping refers to the behavior of adjusting inappropriate impulses and effects consciously, purposefully, and flexibly [17], which is a crucial resource in dealing with distressing circumstances, as it can dramatically diminish the negative effects of stress [18]. According to a study, college students’ strategies for coping with stressful situations were found to be based on their personality traits, and each of the Big Five personalities may predict different coping styles [19]. Further consideration of the relationship between coping and dark personality found that socially aversive personality traits were associated with different coping preferences. For example, Machiavellianism and psychopathy were negatively associated with task-oriented coping, and positively associated with emotionally oriented coping. In contrast, narcissism was found to be associated with task-oriented and emotionally controlled coping [20].

Coping may be another important psychological factor affecting bullying. On the one hand, previous studies have found that bullying is associated with externalizing coping and with a lack of problem-solving [21], that is, bullies tend to adopt negative coping strategies. Misconduct has a significant impact on becoming a bully [22]. On the other hand, some studies have shown that positive coping strategies are effective in reducing bullying behavior in lower grades of elementary school, over time [23]. Specifically, positive coping strategies such as direct coping and active coping can decrease the association between stimuli and bullying, producing the “buffer effect”, and emotion-focused coping strategies amplify the relationship between conflict and bullying [24,25]. It can be seen that coping styles may influence individual bullying behavior.

Gender differences in bullying are another issue that should not be ignored. Norton [26] explored aggressive behavior based on different scales, and the results all showed that boys are more aggressive than girls. More studies have also found that the rate of bullying or participation in bullying is significantly higher for boys than for girls, and this difference is consistent across cultures [27,28]. Sideridis and Alghamdi [29] examined differences in bullying behaviors across gender, finding that boys were more likely to be perpetrators of forms of direct bullying, such as physical bullying. In addition, studies have found that there are gender differences in the Dark Triad [30]. On the other hand, studies have shown that for individuals who have been bullied, positive coping styles such as pro-social behavior may be reduced; however, only girls reported more emotional problems at follow-up [31]. This suggests that there may be gender differences in the psychological mechanisms of bullying.

In summary, it is evident that dark personality traits may positively influence bullying behavior through the mediating effect of coping. At the same time, since a gender difference in bullying between males and females has been relatively consistently found and concluded, whether a gender difference exists in the whole mechanism is worth further discussion. Therefore, we will use a multiple mediation model to explore the mediating effect of coping styles on bullying, and then examine the differences in the mechanism between male and female groups.

Bullying is defined as aggressive goal-directed behavior [32], or is directly described as aggressive behavior [33]. Currently, bullying is mainly measured using aggression measurement instruments. The Buss–Perry Aggression Questionnaire (BPAQ) and Reactive–Proactive Aggression Questionnaire (RPQ) are the most popular instruments for measuring aggressive behavior. The former mainly examines the sub-characteristics of bullying, such as physical aggression, verbal aggression, anger, and hostility [34], while the RPQ divides aggression into reactive aggression and proactive aggression based on the causes of adolescent aggression [35]. Although BPAQ and RPQ differ in theoretical structure, there are still different degrees of correlation in different dimensions [36]. This suggests that we can explore the psychological mechanism of bullying from different perspectives of aggression. If a psychological mechanism of bullying exists in both males and females, no matter what kind of aggressive behavior measurement instrument is used, we should be able to verify this mechanism, and achieve repeatability of the research [37]. Therefore, in the first part of the study, the BPAQ will be used to discuss gender differences in the psychological mechanism of bullying. In the second part of the study, the RPQ will be used to further verify the mechanism and provide more evidence of validity.

Our hypothesis is that there is a mediating effect of coping styles on the relationship between dark personality traits and bullying;Another hypothesis is that there are differences in this mechanism between male and female groups;In the second part of the study, we employed other questionnaires to explore the replicability of this psychological mechanism.

## 2. Methods

### 2.1. Participants

Participants were a convenience sample of 772 students from a high school in Tianjin. Among them, 414 were female (53.600%), and 358 were male (46.400%), with an age range of 14 to 19 years (*M* = 15.860, *SD* = 0.680).

### 2.2. Measures

#### 2.2.1. Dark Personality Traits

The Dirty Dozen (DD) developed by Jonason and Webster [38] was used to administer the test, and the Chinese version was revised by Geng et al. [30]. The questionnaire has 12 items and is scored on a 7-point scale, including the three dimensions of Machiavellianism, psychopathy, and narcissism. The confirmatory factor analysis results showed that the three-factor model data fit well with *X*^2^*/df* = 5.060, CFI = 0.940, TCL = 0.920, SRMR = 0.055, RMSEA = 0.073, and the 90% confidence interval of RMSEA was [0.064, 0.082]. The Cronbach’s alpha for the total scale was 0.847 (0.836 for Machiavellianism, 0.565 for psychopathy, and 0.816 for narcissism).

#### 2.2.2. Bullying

Two different assessment instruments will be used to measure bullying in this study. The first one is the revised Chinese version of the Buss–Perry Aggression Questionnaire (BPAQ) developed by Bryant and Smith in 2001 [39], which consists of ten items, such as “My friends say that I am somewhat argumentative.” Participants rated each BPAQ item using a 5-point scale, ranging from extremely uncharacteristic of me (1) to extremely characteristic of me (5). The Cronbach’s alpha for the scale was 0.779.

The second is the simplified version of the Reactive–Proactive Aggression Questionnaire (RPQ) revised by Mai, Wang, and Liu [40], which includes the two dimensions of reactive and proactive aggression. There were thirteen 3-point items in this scale (1 = never, 2 = sometimes, 3 = often). The results of the validated factor analysis showed that the two-factor model fitted well with *X*^2^*/df* = 3.955, CFI = 0.945, TCL = 0.930, SRMR = 0.042, RMSEA = 0.062, and the 90% confidence interval for RMSEA was [0.053, 0.072]. The Cronbach’s alpha for the total scale was 0.771 (0.786 for reactive aggression, 0.773 for proactive aggression). 

#### 2.2.3. Coping Style

The Simplified Coping Style Questionnaire (SCSQ) used in this study was revised by Xie [41]. The questionnaire contains 20 items with two dimensions, 12 items for positive coping styles (e.g., “Relieve stress through work, study, or other activities”) and 8 items for negative coping styles (e.g., “Trying to rest or take a vacation, temporarily setting aside the problems”). A 4-point scale was used, ranging from do not adopt (1) to often adopt (4). The Cronbach’s alpha for the total scale was 0.736 (0.741 for positive coping styles, 0.669 for negative coping styles).

### 2.3. Data Analysis

The series of studies by Ryu [42] found that multi-group mediation models allow for the comparison of coefficients and effects. Therefore, we used multi-group mediation models to compare the differences in mediation mechanisms between males and females. IBM SPSS 26.0 was used to preprocess data and produce descriptive statistics. Mplus 8.3 was used to construct multi-group mediation models.

## 3. Results

### 3.1. Common Method Bias Test

In this study, the common method bias was controlled by changing the instructions appropriately and using four scoring methods. Harman’s single-factor test was used to test common method bias, and an unrotated exploratory factor analysis was performed for all variables. The results showed that the first factor explained a variance of only 15.536%, which was less than the critical value of 40%, indicating that there was no common method bias.

### 3.2. Gender Differences

To observe the gender differences among the study variables, an independent sample t-test was conducted, and the results are shown in Table 1. The results showed significant differences between males and females in the Dark Triad, Machiavellianism, psychopathy, bullying, and proactive aggression. According to the results, males scored significantly higher than females on Machiavellianism, *t*(759) = −4.344, *p* < 0.001, 95% confidence interval [−2.300, −0.868], *Cohen’s d* = 0.316, and males also scored significantly higher than females on psychopathy, *t*(757) = −3.288, *p* < 0.05, 95% confidence interval [−1.656, −0.418], *Cohen’s d* = 0.239. As expected, males had higher bullying scores compared to females, *t*(746) = −2.335, *p* < 0.05, *Cohen’s d* = 0.171, and males had higher proactive aggression scores compared to females, *t*(758) = −3.392, *p* < 0.001, *Cohen’s d* = 0.247.

### 3.3. Correlation Analysis

A correlation analysis was conducted, and the correlation matrices of all variables are shown in Table 2.

The results indicated that positive coping styles were negatively correlated with bullying, the Dark Triad, Machiavellianism, and psychopathy, but were positively correlated with narcissism. The remaining variables were positively correlated to each other, and the magnitude and direction of the correlation coefficients were as expected.

### 3.4. A Multiple Mediation Model of Bullying

The mediating effects of the negative and positive coping styles were tested using Mplus and the bootstrap method with 5000 bootstrap repetitions; the results are shown in Figure 1. We found that the Dark Triad was positively correlated with bullying (*β* = 0.493, *p* < 0. 001, 95% CI [0.430, 0.556]) and negative coping styles (*β* = 0.236, *p* < 0.001, 95% CI [0.159, 0.313]). However, we found no significant relationship between the Dark Triad and positive coping styles. The positive effect of negative coping styles on bullying was significant (*β* = 0.173, *p* < 0.001, 95% CI [0.097, 0.248]), and there was no significant relationship between positive coping styles and bullying. The indirect effect of negative coping styles was 0.041m with a boot CI of [0.019, 0.063]. The indirect effect of negative coping styles was 0.005 (boot CI [−0.003, 0.012]), and the total effect was 0.538, with a boot CI of [0.482, 0.595]. The mediating effect size of negative coping styles was 7.621%.

### 3.5. Multi-Group Multiple Mediation Models of Bullying

We divided males and females into two groups and conducted an analysis to explore the mediating effects separately. The results showed that the partial mediating effect of negative coping styles differed from that of positive coping styles (Figure 2).

For the female group, the Dark Triad was positively correlated with bullying (*β* = 0.504, *p* < 0.001, 95% CI [0.427, 0.581]) and negative coping styles (*β* = 0.206, *p* < 0.001, 95% CI [0.059, 0.245]), but was negatively correlated with positive coping styles (*β* = −0.203, *p* < 0.001, 95% CI [−0.189, −0.028]). The indirect effect of negative coping styles was 0.031 (boot CI [0.007, 0.055]), the indirect effect of positive coping was 0.022 (boot CI [0.003, 0.041]), and the total effect was 0.557, with a boot CI of [0.489, 0.626]. The mediating effect sizes of negative and positive coping styles were 5.566% and 3.950%.

The results of the male group indicated that the Dark Triad was positively correlated with bullying (*β* = 0.460, *p* < 0.001, 95% CI [0.356, 0.564]) and negative coping styles (*β* = 0.254, *p* < 0.001, 95% CI [0.137, 0.371]). In addition, the positive effect of negative coping styles on bullying was significant (*β* = 0.191, *p* < 0.01, 95% CI [0.076, 0.307]). However, we found no significant relationship between the Dark Triad and positive coping styles, and no significant relationship between positive coping styles and bullying. The indirect effect of negative coping styles was 0.049 (boot CI [0.010, 0.087]), the indirect effect of positive coping was 0.000 (boot CI [−0.010, 0.009]), and the total effect was 0.509, with a boot CI of [0.418, 0.598]. The mediating effect size of negative coping styles was 9.627%.

### 3.6. Multi-Group Multiple Mediation Models of Proactive Aggression

To validate the gender differences once again, we incorporated the results of the Reactive–Active Aggression Scale into the model. It transpired that the partial mediating effect of negative and positive coping styles was found to differ between genders (Figure 3). 

For the female group, the Dark Triad was positively correlated with proactive aggression (*β* = 0.286, *p* < 0.001, 95% CI [0.180, 0.382]) and negative coping styles (*β* = 0.210, *p* < 0.001, 95% CI [0.114, 0.306]), but was negatively correlated with positive coping styles (*β* = −0.205, *p* < 0.001, 95% CI [−0.299, −0.111]). The positive effect of negative coping styles on proactive aggression was significant (*β* = 0.171, *p* < 0.01, 95% CI [0.055, 0.288]), and the negative effect of positive coping styles on proactive aggression was also significant (*β* = −0.125, *p* < 0.01, 95% CI [−0.217, −0.033]). The indirect effect of negative coping styles was 0.036 (boot CI [0.005, 0.067]), the indirect effect of positive coping was 0.026 (boot CI [0.002, 0.049]), and the total effect was 0.347, with a boot CI of [0.254, 0.440]. The mediated effect sizes of negative and positive coping styles were 10.374% and 7.493%.

The results of the male group indicated that the positive predictive effect of the Dark Triad on proactive aggression was significant *(β* = 0.234, *p* < 0.001, 95% CI [0.118, 0.350]), the positive effect of the Dark Triad on negative coping styles was significant (*β* = 0.253, *p* < 0.001, 95% CI [0.135, 0.371]), and the positive effect of negative coping styles on proactive aggression was significant (*β* = 0.213, *p* < 0.001, 95% CI [0.096, 0.330]). Furthermore, the negative effects of the Dark Triad on positive coping styles and positive coping styles on proactive aggression were not significant. The indirect effect of negative coping styles was 0.054 (boot CI [0.015, 0.093]), the indirect effect of positive coping was 0.0001 (boot CI [−0.007, 0.009]), and the total effect was 0.289, with a boot CI of [0.184, 0.394]. The mediated effect size of negative coping styles was 18.685%. 

## 4. Discussion

This study focused on the issue of the psychological mechanisms of bullying and their gender differences, while introducing proactive aggression for further validation. A T-test was used to compare the gender differences among the study variables. Then, we explored the mediating role of coping style, and further explored the psychological mechanisms of bullying by dividing the participants into two groups based on gender. We found that there were gender differences in their psychological mechanisms. To verify this result, we replaced bullying with proactive aggression for validation, and the results showed that the gender differences in psychological mechanisms still existed.

We found there was a gender difference in bullying and proactive aggression, with male scores significantly higher than female scores, while there was no significant gender difference in reactive aggression. This result may be due to the differences in the developmental process of reactive–proactive aggression between males and females, i.e., in childhood, there is no gender difference in reactive and proactive aggression; in adolescence, there is a gender difference in proactive aggression, but not in reactive aggression; and in adulthood, there is a gender difference in both dimensions of reactive–proactive aggression [43,44]. Specifically, during adolescence, there are gender differences in proactive aggression but not reactive aggression [44,45]. This may be attributed to the fact that boys experience an increase in height and weight during adolescence [46], providing them with a physical advantage that makes proactive aggression a viable strategy. Consequently, a gender difference in proactive aggression emerges. However, as the prefrontal cortex, which regulates impulsive behavior, undergoes maturation during adolescence [47], reactive aggression does not show the same increase with age. Therefore, there is no significant difference in reactive aggression between adolescent boys and girls. In adulthood, gender differences are observed in both proactive and reactive aggression. This could be attributed to the different gender roles prevalent in Chinese culture, which expect females to exhibit traits such as politeness, gentleness, and modesty. As a result, females may be less likely to engage in aggressive behaviors compared to males. However, the issue of gender differences in proactive and reactive aggression during childhood remains unclear, and warrants further research for better understanding.

The results of the t-test for the independent variable of the Dark Triad showed that there were gender differences in Machiavellianism and psychopathy, with males scoring significantly more than females, while there were no gender differences in narcissism, which is the same as the results of other studies. Moreover, compared with females, males showed higher correlation coefficients between Machiavellianism and aggression, and psychopathy and aggression, indicating a stronger and closer relationship [30].

The gender differences in narcissism are not significant, and it is worth thinking more about the substance of narcissism. Narcissism refers to a pathological form of self-love whose psychological behavior is characterized by exaggeration, conceit, arrogance, etc. [48]. Rauthmann and Kolar [49] asked participants to evaluate the Dark Triad in terms of desirability, consequences for the self, and consequences for others. They found that narcissism was evaluated as “brighter” than Machiavellianism and psychopathy in lay people’s perceptions. In a 2022 study, narcissism was also defined by researchers as a “brighter” trait [50]. Our study corroborated their findings; specifically, positive coping styles were negatively correlated with Machiavellianism and psychopathy and positively correlated with narcissism, but negative coping styles were positively correlated with Machiavellianism, psychopathy, and narcissism.

We found that the Dark Triad can influence bullying through the mediating effect of negative coping styles, but not positive coping styles. When the participants were further divided into male and female groups, their psychological mechanisms were found to be different, with the Dark Triad acting on bullying through negative and positive coping styles in the female group; meanwhile, in the male group, the Dark Triad only acted on bullying through negative coping styles. We replaced bullying with aggression to validate the result, and found that differences in psychological mechanisms persisted. Differences between the male and female groups in the mediation model were estimated in the modeling of the multi-group model, and were therefore comparable. 

Mark and Smith [51] proved that negative coping is more strongly associated with mental health issues than positive coping. It is also clear from the results of our study that negative coping styles were more strongly associated with bullying and aggression than positive coping styles. In the educational environment of China, teenagers are used to remaining silent in the presence of stressful life events. Reducing such negative coping styles in the developmental path of teenagers is more conducive to reducing adolescent bullying and promoting physical and mental health. In our study, there were different mediating effect results for positive coping styles among the male and female groups, which may reflect the differences in the buffering hypothesis presented by different individuals [52]. There may be differences in the buffer effect of positive coping styles between the male and female groups. Specifically, the mediating effect of negative coping styles was found in both male and female groups, whereas the mediating effect of positive coping styles was found only in the female group. The results of our study indicated that the destructive effect of negative coping styles was much greater than the facilitative effect of positive coping styles.

In the first part of the study, the mediating effect size of coping style was not high, indicating partial mediation. The other factors that influence bullying and the interaction of other causes need to be explored. In addition, the study explored “why” the Dark Triad would lead to bullying, and further explored the effect of the Dark Triad on bullying in different conditions by dividing males and females into two groups. The predictive relationship between the Dark Triad and bullying under other conditions is worth further exploration.

The results of the second part of the study were validated using a simplified version of the Reactive–Proactive Aggression Questionnaire, replacing bullying with proactive aggression; the results were consistent with the second part of the study in that the difference in psychological mechanisms still existed. However, there are still some differences between aggression and bullying, and the two measurement instruments have different theoretical backgrounds. It can also be seen from the correlation analysis results that reactive and proactive aggression are significantly different in the degree of correlation between bullying and the Dark Triad. Because there are inherent differences between reactive and proactive aggression, it is necessary to distinguish between them. The social information processing model (SIP), mentioned by Crick and Dodge [53], suggests that proactive aggressors have higher self-efficacy for aggression, and in terms of evaluation of aggressive behaviors, proactive aggressors believe that aggression brings more positive results. The specific differences between reactive and proactive aggression deserve further exploration.

## 5. Limitations and Implications

In our study, we not only found that there were gender differences in Machiavellianism, psychopathy, bullying, and aggression, but also found that there were gender differences in the mediating effects of coping styles on the mechanisms of action of the Dark Triad, bullying, and proactive aggression. However, it is important to acknowledge several limitations inherent in our investigation.

Firstly, due to the cross-sectional design of our study, we are unable to establish definitive causal relationships. To further elucidate the formation and underlying mechanisms of bullying behavior, future research endeavors should employ longitudinal tracking studies and behavioral experiments.

Furthermore, our study has elucidated the mediating role of coping styles in the association between dark personality traits and bullying behavior. Additionally, we have observed gender disparities in the psychological mechanisms underlying these relationships. These findings offer valuable insights into the nature of aggressive behavior, highlighting the existence of distinct psychological mechanisms. Consequently, our results contribute to our shared understanding of bullying behavior, and inform the development of targeted interventions for middle school students. To comprehensively address the issue of bullying, it is imperative to adopt a multidisciplinary approach and consider multiple perspectives, facilitating a comprehensive understanding of the phenomenon and the implementation of effective interventions. Lastly, our discussion emphasizes the heightened association between negative coping strategies and both bullying and aggression, underscoring the importance of addressing maladaptive coping patterns in intervention programs.

In conclusion, despite the limitations mentioned, our study provides valuable insights into the complex interplay between dark personality traits, coping styles, and bullying behavior. Future research should continue to explore these relationships using more rigorous methodologies, ultimately enhancing our understanding of bullying and paving the way for effective prevention and intervention strategies.

## Figures and Tables

**Figure 1 behavsci-13-00532-f001:**
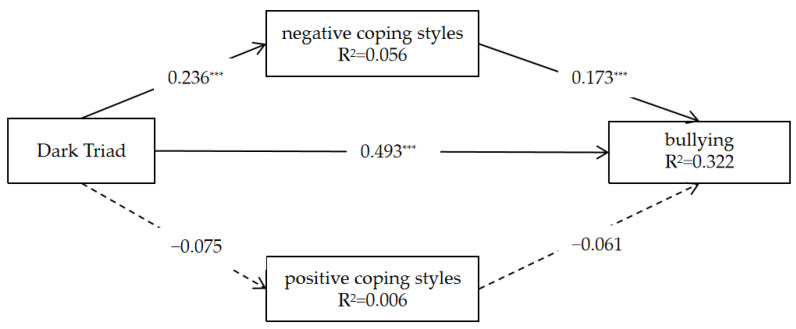
The multiple mediation model with standardized estimates. *** *p* < 0.001.

**Figure 2 behavsci-13-00532-f002:**
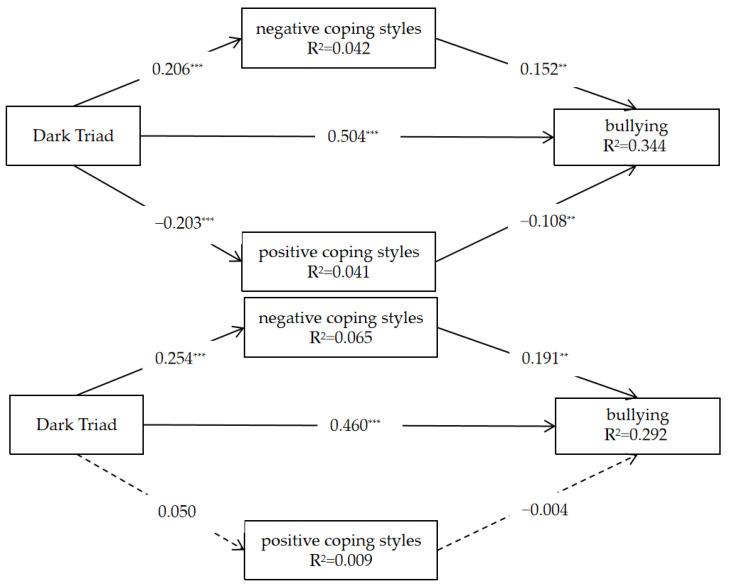
Results of the mediating effect test. (**Above**) is the female group, and (**below**) is the male group.*** *p* < 0.001; ** *p* < 0.01.

**Figure 3 behavsci-13-00532-f003:**
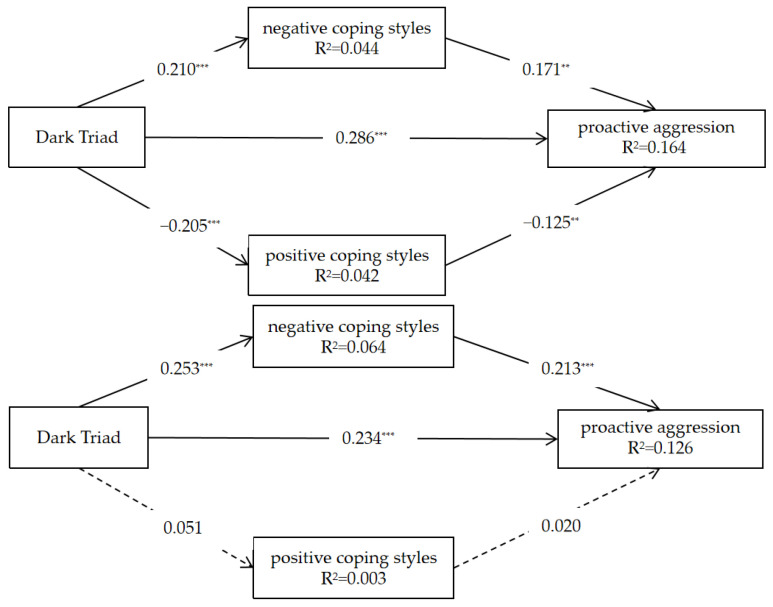
Results of the mediating effect test. (**Above**) is the female group, and (**below**) is the male group.*** *p* < 0.001; ** *p* < 0.01.

**Table 1 behavsci-13-00532-t001:** Gender differences in all variables.

	*M_male_* (*SD*)	*M_female_* (*SD*)	*t*
Bullying	25.474 (7.115)	24.270 (6.954)	−2.335 *
Dark Triad	35.171 (12.621)	31.993 (11.887)	−3.539 ***
Dark Triad, Machiavellianism	9.474 (5.450)	7.891 (4.608)	−4.344 ***
Dark Triad, psychopathy	10.157 (4.482)	9.120 (4.196)	−3.288 **
Dark Triad, narcissism	15.637 (5.785)	15.125 (5.982)	−1.198
Positive coping styles	21.602 (5.634)	21.359 (5.222)	−0.606
Negative coping styles	9.814 (3.782)	10.396 (4.370)	−1.867
Aggression behavior	9.008 (2.467)	9.152 (2.375)	0.818
Proactive aggression	6.847 (1.669)	6.500 (1.128)	−3.392 ***
Reactive aggression	15.863 (3.357)	15.654 (2.867)	−0.927

Note: *** *p* < 0.001; ** *p* < 0.01;* *p* < 0.05.

**Table 2 behavsci-13-00532-t002:** Correlation matrix.

Variables	1	2	3	4	5	6	7	8	9
1. Bullying	1								
2. Dark Triad	0.541 ***	1							
3. Dark Triad, Machiavellianism	0.426 ***	0.851 ***	1						
4. Dark Triad, psychopathy	0.450 ***	0.769 ***	0.613 ***	1					
5. Dark Triad, narcissism	0.437 ***	0.789 ***	0.464 ***	0.335 ***	1				
6. Positive coping styles	−0.084 *	−0.075 *	−0.114 **	−0.154 ***	0.053	1			
7. Negative coping styles	0.270 ***	0.238 ***	0.190 **	0.182 ***	0.190 ***	0.184 ***	1		
8. Proactive aggression	0.274 ***	0.319 ***	0.346 **	0.272 **	0.171 ***	−0.012	0.262 ***	1	
9. Reactive aggression	0.558 ***	0.332 ***	0.210 ***	0.246 ***	0.328 ***	−0.041	0.198 ***	0.258 ***	1

Note: *** *p* < 0.001; ** *p* < 0.01;* *p* < 0.05.

## Data Availability

The data presented in this study are available on request from the corresponding author.

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
