# Peer review of "The Role of Coping Styles in Mediating the Dark Triad and Bullying: An Analysis of Gender Difference"

_behavsci, 2023, doi:10.3390/bs13070532_

Round 1
Reviewer 1 Report
The Review of Research Article
“Multiple Mediation of Coping Styles in the Relationship Between the Dark Triad and Bullying: An Analysis of Gender Differences”
During the analysis of the materials of this study, presented in the article “ “Multiple Mediation of Coping Styles in the Relationship Between the Dark Triad and Bullying: An Analysis of Gender Differences”, I come to the conclusion that the article title and abstract are appropriate.
The purpose of the article and its significance is stated clearly. The study methods are sound and appropriate. The writing is clear and concise. The conclusions are accurate and supported by the content. The article is of interest to members of the clinical and neurophysiological research community. The study was conducted according to the guidelines of the Declaration of Helsinki, and approved by the Regional Ethics Review Board.
I recommend Research Article “Multiple Mediation of Coping Styles in the Relationship Between the Dark Triad and Bullying: An Analysis of Gender Differences” for publication.
Author Response
Thank you very much for taking the time to read our research article "Multiple Mediators of Coping Style in the Relationship Between the Dark Triad and Bullying: An Analysis of Gender Differences". We appreciate your positive comments on the appropriateness of the title and abstract, the clarity and conciseness of the writing, the accuracy of the conclusions, and the soundness and appropriateness of our research methodology.
We thank you for recommending our research articles for publication, and we appreciate your thoughtful feedback.
Reviewer 2 Report
This perspective about the Dark Triad personality and bullying is interesting. Nevertheless, there are also some aspects to improve.
Line 1: The title is complicated. Try to do it easier according to the paper's purpose.
Lines 15 and 31: The text Is not clear about its purpose. First, it seems to be to reduce the occurrence of bullying, but the end spots towards the significance of gender differences. Gender is a characteristic of the individuals, but a predictor of bullying? Just based on sex? And what about the relationship between DT and bullying? There is a lack of coherence among these sentences.
Line 75: This sentence seems to be unnecessary.
Lines 101-118: This is not the place for conclusions. You should describe here the independent and dependent variables, hypothesis, or a summary of research after you show all the concepts. However, line 109 starts with the definition of bullying.
Line 126: Rephrase points 1 and 2 as separate hypotheses. It seems to be incomplete sentences.
Line 166: Examples of items are necessary to make conclusions more understandable.
Line 199: Capital letters.
Lines 235 and 264: This information must be in the figure description or inside the figure.
Line 275: The limitations of the study should be in the discussion. Where are they? There is also a lack of explanation about the differences of gender and the types of bullying.
Line 352: The conclusion is too short. The focus of your paper is on Dark Triad personality and bullying. There is a lack of ideas or proposals about how your findings help to identify, prevent or avoid bullying.
Line 370: It would be necessary to increase the recent references about gender and bullying.
Author Response
We greatly appreciate your valuable feedback. Based on your comments, we have made careful revisions to the manuscript. The specific modifications are outlined below, with changes in the article highlighted in blue font.
This perspective about the Dark Triad personality and bullying is interesting. Nevertheless, there are also some aspects to improve.
Q1: Line 1: The title is complicated. Try to do it easier according to the paper's purpose.
Thanks to the reviewers for your correction. Based on your feedback, we revised the title from “Multiple Mediation of Coping Styles in the Relationship Between the Dark Traid and Bullying: An Analysis of Gender Differences” to “The Role of Coping Styles in Mediating Dark Triad and Bullying: An Analysis of Gender Difference” to better reflect the research purpose.
Q2: Lines 15 and 31: The text Is not clear about its purpose. First, it seems to be to reduce the occurrence of bullying, but the end spots towards the significance of gender differences. Gender is a characteristic of the individuals, but a predictor of bullying? Just based on sex? And what about the relationship between DT and bullying? There is a lack of coherence among these sentences.
Thanks to the reviewers for your correction. We apologize for any confusion caused by the lack of clarity in the text. The purpose of the paragraph is twofold: to examine the mediating effect of coping styles on the relationship between dark personality traits (DT) and bullying, and to explore gender differences in this mechanism.
Regarding gender as a predictor of bullying, it is noteworthy that the study revealed significant gender differences in bullying scores, with males scoring significantly higher than females. This suggests that there may be a gender disparity in bullying behavior. However, it is important to consider that gender alone may not be the sole predictor of bullying, and other factors could also contribute to this behavior.
In relation to the association between DT and bullying, the paragraph discusses the utilization of mediation models in the study to examine the potential mediating effect of coping styles on this relationship. Specifically, coping styles were investigated as a potential mechanism through which DT influences bullying behavior. The findings revealed that negative coping styles partially mediated the relationship between DT and bullying in both males and females, whereas positive coping styles only served as a mediator in females.
Overall, the paragraph aims to present key findings related to the mediation of coping styles and gender differences in the relationship between DT and bullying. However, we have made revisions to the abstract to address any unclear or incoherent aspects and ensure a clearer presentation of the research purpose and findings.
For example, the sentence "Gender was found to significantly predict bullying, with males scoring higher than females" can be modified to "Significant gender differences in bullying behavior were observed, with males scoring significantly higher than females."
Q3: Line 75: This sentence seems to be unnecessary.
Thanks to the reviewers for your correction. We have removed the sentence “For this reason, dark personality traits may also be predictive of certain coping styles.”
Q4: Lines 101-118: This is not the place for conclusions. You should describe here the independent and dependent variables, hypothesis, or a summary of research after you show all the concepts. However, line 109 starts with the definition of bullying.
Thanks to the reviewers for your correction.We apologize for any confusion caused by our wording. We understand that this section is not meant to draw conclusions but rather to provide a summary and clarification of the previous content. To avoid any further misunderstanding, we have revised this paragraph accordingly. “In summary, it is evident that dark personality may positively influence bullying behavior through the mediating effect of coping.”
Q5: Line 126: Rephrase points 1 and 2 as separate hypotheses. It seems to be incomplete sentences.
Thanks to the reviewers for your correction. Upon reviewing the provided suggestions, we have made the following revisions:
Rephrase point 1 as a hypothesis: "Our hypothesis is that there is a mediating effect of coping styles on the relationship between dark personality traits and bullying."
Rephrase point 2 as a separate hypothesis: "Another hypothesis is that there are differences in this mechanism between male and female groups."
Q6: Line 166: Examples of items are necessary to make conclusions more understandable.
Thanks to the reviewers for your correction. To better understand the conclusion, we have listed the relevant questionnaire items for positive and negative coping styles separately. ”12 items for positive coping styles(e.g.,” Relieve stress through work, study, or other activities”) and 8 items for negative coping styles(e.g.,” Trying to rest or take a vacation, temporarily setting aside the problems”).”
Q7:Line 199: Capital letters.
Thanks to the reviewers for your correction. We apologize for this oversight.We have capitalized the first letter as suggested.
Q8: Lines 235 and 264: This information must be in the figure description or inside the figure.
Thanks to the reviewers for your correction. We have addressed this oversight and updated the image description to reflect the corresponding information.
Q9: Line 275: The limitations of the study should be in the discussion. Where are they? There is also a lack of explanation about the differences of gender and the types of bullying.
Thanks to the reviewers for your correction. We have addressed the limitations of our study in the fifth section titled "Limitations and Implications." By introducing the section with the statement "However, it is important to acknowledge several limitations inherent in our investigation," we proceeded to elaborate on the specific limitations of our study.
Q10: Line 352: The conclusion is too short. The focus of your paper is on Dark Triad personality and bullying. There is a lack of ideas or proposals about how your findings help to identify, prevent or avoid bullying.
Thanks to the reviewers for your correction. Due to our negligence, we only briefly wrote the conclusion. Now we have expanded it to "Limitations and Implications", including the conclusion, how our findings help to identify, prevent, or avoid bullying, and the limitations and shortcomings of our study.
Q11: Line 370: It would be necessary to increase the recent references about gender and bullying.
Thanks to the reviewers for your correction. We have updated approximately 20% of the references in the article to ensure its currency and reliability.

Reviewer 3 Report
Thank you for giving me the opportunity to review this manuscript. I think it is a very interesting and meaningful study that explores the relationship between the dark traid and bullying, focusing on gender differences. I have some suggestions which could help to improve the present manuscript.
Q1: In lines 48-49, "The occurrence of bullying is influenced by many factors, and personality is one of the important psychological factors." What other psychological factors are there? Why is only the factor of personality selected in this study? Instead of selecting other factors for research, the author is requested to add relevant content.
Q2: In fact, this research wants to explore the relationship between dark triad and these variables, but for lines 48-64, it is more like saying that both the bright side of personality and the dark side of the personality will affect bullying. Therefore, in lines 64-65, "It can be seen from these studies that personality may be an important predictor of bullying", personality should be replaced to dark triad.
Q3: In lines 124-126, The researcher divided the research into Study 1 and Study 2, but the paper was not written according to the content of the two studies. Suggest changing "Therefore, in study 1, the BPAQ will be used to discuss gender differences in the psychological mechanism of bullying. In study 2, the RPQ will be used to further verify the mechanism and provide more evidence of validity". to "This study first use BPAQ to discuss gender differences in the psychological mechanism of bullying. and then using RPQ to further verify the mechanism and provide more evidence of validity".
Q4: In lines 196-198, the correlation between positive coping style and some variables is not significant, but this paper still explains all the positive and negative correlations.
Q5: In lines 209-211, there are two indirect effects of negative coping strategies, one is 0.041 and the other is 0.005. The latter 0.005 should be the indirect effect of positive coping strategies, but later it is mentioned that the indirect effect of positive coping is 0.026. It is suggested that the author check the results of this part.
Q6: Line 231, "The left is the female group, and the right is the male group", but the figures in lines 228 and 229 are displayed vertically instead of horizontally. The same issue occurs in line 260.
Q7: Line 238, "the indirect effect of negative coping styles was 0.049", the direct effect was 0.460, and the total effect should be 0.509, with a mediation effect of 9.627%. However, in line 239, the total effect is reported to be 0.508, with a mediation effect of 9.646%.
None.
Author Response
We greatly appreciate your valuable feedback. Based on your comments, we have made careful revisions to the manuscript. The specific modifications are outlined below, with changes in the article highlighted in green font.
Q1: In lines 48-49, "The occurrence of bullying is influenced by many factors, and personality is one of the important psychological factors." What other psychological factors are there? Why is only the factor of personality selected in this study? Instead of selecting other factors for research, the author is requested to add relevant content.
Thank you for your feedback and suggestion. You are correct that the occurrence of bullying is influenced by multiple psychological factors. In this particular study, our focus was on examining the mediating effect of coping styles on the relationship between dark personality traits and bullying. While personality is indeed an important psychological factor, we acknowledge that there are other factors that can contribute to the occurrence of bullying.
To address your concern and provide a more comprehensive understanding, we will consider expanding the discussion in our article to include relevant content about other psychological factors that are known to influence bullying. This will help provide a more holistic perspective on the subject and enhance the overall quality of the research.
Q2: In fact, this research wants to explore the relationship between dark triad and these variables, but for lines 48-64, it is more like saying that both the bright side of personality and the dark side of the personality will affect bullying. Therefore, in lines 64-65, "It can be seen from these studies that personality may be an important predictor of bullying", personality should be replaced to dark triad.
Thank you for your feedback and suggestion. We appreciate your careful reading of the manuscript. After considering your comment, we agree that the use of the term "personality" in lines 64-65 may not accurately reflect the focus of our research, which specifically investigates the relationship between the dark triad traits and bullying.To address this concern, We will make the necessary revisions in the manuscript as suggested.
Q3: In lines 124-126, The researcher divided the research into Study 1 and Study 2, but the paper was not written according to the content of the two studies. Suggest changing "Therefore, in study 1, the BPAQ will be used to discuss gender differences in the psychological mechanism of bullying. In study 2, the RPQ will be used to further verify the mechanism and provide more evidence of validity". to "This study first use BPAQ to discuss gender differences in the psychological mechanism of bullying. and then using RPQ to further verify the mechanism and provide more evidence of validity".
Thank you for your feedback and suggestion. The intention here is to emphasize the second part of the study, so it was not specifically written following the format of two separate studies. This is because the data were collected together, with only a logical distinction. We have also made modifications to eliminate any ambiguity and clearly label it as the second part of the study.
Q4: In lines 196-198, the correlation between positive coping style and some variables is not significant, but this paper still explains all the positive and negative correlations.
Thank you for your feedback and suggestion. We have removed the correlation results between positive coping styles and proactive aggression and reactive aggression from the description of the relevant results.
Q5: In lines 209-211, there are two indirect effects of negative coping strategies, one is 0.041 and the other is 0.005. The latter 0.005 should be the indirect effect of positive coping strategies, but later it is mentioned that the indirect effect of positive coping is 0.026. It is suggested that the author check the results of this part.
Thank you for your feedback and suggestion. Due to our oversight, the mention of "0.026" in this context was an error and has been removed.
Q6: Line 231, "The left is the female group, and the right is the male group", but the figures in lines 228 and 229 are displayed vertically instead of horizontally. The same issue occurs in line 260.
Thank you for your feedback and suggestion. We apologize for the formatting error in our paper submission. It has been rectified now, and we assure you that such mistakes should not have occurred in the first place. We apologize for any inconvenience caused and assure you that it will not interfere with your review process.
Q7: Line 238, "the indirect effect of negative coping styles was 0.049", the direct effect was 0.460, and the total effect should be 0.509, with a mediation effect of 9.627%. However, in line 239, the total effect is reported to be 0.508, with a mediation effect of 9.646%.
Thank you for your feedback and suggestion. Upon comparing the data results with the research findings, we did find that there were several errors here, for which we are very sorry. Thank you once again for your valuable feedback.

Round 2
Reviewer 2 Report
Line 130: this sentence is repeted.
Line 122: lowercase letter
Line 290: the differences in the developmental process of reactive-proactive aggression between males and females
That differences should be explain. Which factor is acting along the ages? There is a lack of social factors consideration to explain differences between males and females, taking into account that there are no differences in childhood. Or at least, you could consider that factor to be explored for future research.
Line 442: Why capital letters?
Author Response
We greatly appreciate your valuable feedback. Based on your comments, we have made careful revisions to the manuscript. The specific modifications are outlined below, with highlight changes in the article in edit mode.
Q1: Line 130: this sentence is repeted.
Thank you for your careful review. We apologize for the oversight that led to this error. We have now revised the passage to read as follows: "Finally, we employed other questionnaires to explore the replicability of this psychological mechanism."
Q2: Line 122: lowercase letter
Thank you for bringing the oversight to our attention. We apologize for neglecting to capitalize certain words during the sentence segmentation. We have now made the necessary corrections to ensure proper capitalization throughout the manuscript.
Q3: Line 290: the differences in the developmental process of reactive-proactive aggression between males and females
That differences should be explain. Which factor is acting along the ages? There is a lack of social factors consideration to explain differences between males and females, taking into account that there are no differences in childhood. Or at least, you could consider that factor to be explored for future research.
Thank you very much for the feedback on the review. We have made a supplement to this in line 334 of the manuscript, providing corresponding explanations from both physiological and social perspectives.
“Specifically, during adolescence, there are gender differences in proactive aggression but not reactive aggression [44,45]. This may be attributed to the fact that boys experience an increase in height and weight during adolescence [46], providing them with a physical advantage that makes proactive aggression a viable strategy. Consequently, a gender difference in proactive aggression emerges. However, as the prefrontal cortex, which regulates impulsive behavior, undergoes maturation during adolescence [47], reactive aggression does not show the same increase with age. Therefore, there is no significant difference in reactive aggression between adolescent boys and girls. In adulthood, gender differences are observed in both proactive and reactive aggression. This could be attributed to the different gender roles prevalent in Chinese culture, which expect females to exhibit traits such as politeness, gentleness, and modesty. As a result, females may be less likely to engage in aggressive behaviors compared to males. However, the issue of gender differences in proactive and reactive aggression during childhood remains unclear and warrants further research for better understanding.”
Q4: Line 442: Why capital letters?
Thank you for pointing out our oversight in the citation of the reference. We apologize for the mistake and have made the necessary revisions. Additionally, we have carefully checked and verified the citation format for all other references in the manuscript.
